# G²RPO: Granular GRPO for Precise Reward in Flow Models

## Abstract

The integration of online reinforcement learning (RL) into diffusion and flow models has recently emerged as a promising approach for aligning generative models with human preferences. Stochastic sampling via Stochastic Differential Equations (SDE) is employed during the denoising process to generate diverse denoising directions for RL exploration. While existing methods effectively explore potential high-value samples, they suffer from sub-optimal preference alignment due to sparse and narrow reward signals. To address these challenges, we propose a novel **G**ranular-**GRPO** (G²RPO ) framework that achieves precise and comprehensive reward assessments of sampling directions in reinforcement learning of flow models. Specifically, a *Singular Stochastic Sampling* strategy is introduced to support step-wise stochastic exploration while enforcing a high correlation between the reward and the injected noise, thereby facilitating a faithful reward for each SDE perturbation. Concurrently, to eliminate the bias inherent in fixed-granularity denoising, we introduce a *Multi-Granularity Advantage Integration* module that aggregates advantages computed at multiple diffusion scales, producing a more comprehensive and robust evaluation of the sampling directions. Experiments conducted on various reward models, including both in-domain and out-of-domain evaluations, demonstrate that our G²RPO significantly outperforms existing flow-based GRPO baselines, highlighting its effectiveness and robustness.

## 1 Introduction

Recent advances in generative models, particularly diffusion models (Ho et al., 2020; Song et al., 2020a;b) and flow models (Lipman et al., 2022; Liu et al., 2022; Peebles & Xie, 2023), have revolutionized visual content creation, offering unprecedented capabilities in generating high-quality images (Rombach et al., 2022; Podell et al., 2023; Esser et al., 2024; Labs, 2024) and videos (Blattmann et al., 2023; Chen et al., 2024; Guo et al., 2023; Kong et al., 2024; Wan et al., 2025). However, a key challenge remains in aligning model outputs with the diverse and complex human preferences. To tackle this challenge, reinforcement learning from human feedback (RLHF) (Fan et al., 2023; Black et al., 2023) has emerged as a promising solution, characterized by its adaptability and cost-effectiveness. Paradigms such as Proximal Policy Optimization (PPO) (Schulman et al., 2017), Direct Policy Optimization (DPO) (Rafailov et al., 2023), and Group Relative Policy Optimization (GRPO) (Shao et al., 2024) have been introduced. Among these, GRPO stands out as an innovative online reinforcement learning approach. By leveraging group comparisons to optimize policies, GRPO eliminates the need for a separate value model, achieving greater flexibility and scalability.

To integrate GRPO into flow-based generative models, Flow-GRPO (Liu et al., 2025) and Dance-GRPO (Xue et al., 2025) substitute the deterministic ODE sampler with an SDE formulation, wherein the injected stochasticity deliberately perturbs the denoising direction at each step. Although the resulting samples enable exhaustive per-step exploration for reinforcement learning, they simultaneously underscore the difficulty of attributing the final reward to any specific random perturbation, thereby constraining model trainability. Specifically, most existing flow-based GRPO methods encounter two core issues in evaluating group denoising directions: 1) **Sparse reward**: As shown in Fig. 1 (b), the final reward signal is uniformly assigned to each SDE sampling step, which cannot be precisely aligned with the sampling direction at each step, leading to inaccuracies for the optimization at individual steps. 2) **Incomplete evaluation**: As depicted in Fig. 1 (c), each denois-

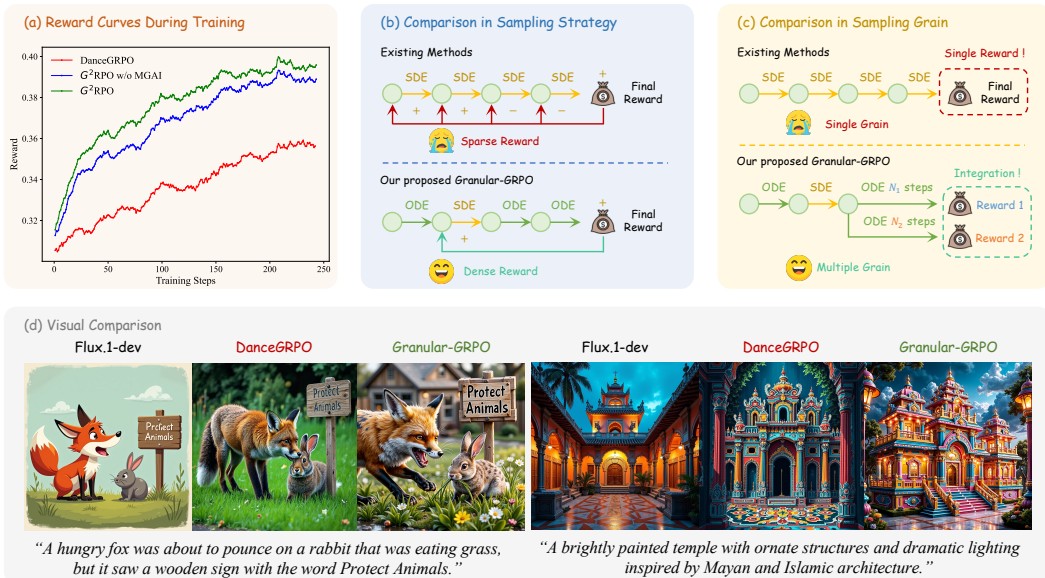

Figure 1: **Comparison between our G²RPO and existing studies**. (a) G²RPO significantly outperforms DanceGRPO in reward scores (HPS-v2.1 in this figure). (b) Sampling strategy comparison. G²RPO acquire a dense reward by confining stochasticity to individual sampling steps. (c) Sampling grain comparison. G²RPO achieves a comprehensive evaluation of each sampling direction by integrating advantages from multi-granularity ODE denoising. (d) Visual Comparison. Compared to the baseline method, the images generated by G²RPO are more aligned with human preferences.

ing direction is bound to a fixed number of denoising steps, resulting in a singular granularity of denoised images, which impairs the reward model's ability to conduct a comprehensive comparison across the group.

To address these limitations, we propose **G**ranular-**GRPO** (G²RPO ) , a novel online reinforcement learning framework specifically designed for precise and comprehensive reward signals. First, mirroring the sparse-reward problem (Hare, 2019; Liang et al., 2024) that plagues RLHF, the reward signal in the SDE sampling process is delivered only after an entire sequence of decisions. This long delay undermines the credit-assignment chain, preventing the linking of the terminal reward with any specific earlier action and thereby inducing sluggish, unstable learning. Therefore, we propose a simple yet effective sampling strategy, termed **Singular Stochastic Sampling**. As illustrated in Fig 1 (b), this strategy applies the SDE formulation at a single time step to generate a group of denoising directions, while employing deterministic ODE sampling for all other steps. By concentrating stochasticity at one specific step, the proposed method establishes a strong correlation between the reward signal and the injected noise, enabling stable model optimization. Secondly, we propose a **Multi-Granularity Advantage Integration** (MGAI) module. As depicted in Fig. 1(c), instead of binding each denoising direction to a fixed subsequent denoising granularity, the denoising directions in the same group are assigned to a spectrum of denoising steps, producing images with different granularities. The corresponding reward signals of these images are then fused into a unified advantage estimate, yielding a comprehensive evaluation of the current state's value.

With the support of the Singular Stochastic Sampling strategy and the Multi-Granularity Advantage Integration module, G²RPO can provide a more precise and comprehensive reward signal, thereby enhancing the upper limit of the GRPO model training. As shown in Fig 1 (a), our reward curves exhibit stable and significant improvements over the baseline during training. Additionally, Fig 1 (d) illustrates the images generated by G²RPO , highlighting its advantages in text prompt adherence and detail fidelity.

Our contributions can be summarized as follows: (1) **Granular-GRPO**: A novel flow-based GRPO framework designed to provide a precise and comprehensive evaluation of the denoising directions sampled by the SDE, thereby improving the precision of model optimization. (2) **Singular Stochastic Sampling**: A sampling strategy confines stochasticity to individual sampling steps, addressing the sparse reward issue associated with long-range stochasticity injection. (3) **Multi-Granularity**

**Advantage Integration**: A module integrates the advantages of multi-granularity denoised images and enables a comprehensive evaluation of each sampling direction. (4) **Superior Performance**: Extensive experiments across various reward models demonstrate that our G$^2$RPO significantly outperforms existing baselines, demonstrating its effectiveness and robustness.

## 2 RELATED WORK

**Alignment for Large Language Models**. Recent years have witnessed a paradigm shift from supervised fine-tuning (Dong et al., 2023; Sun, 2024) to multi-turn online reinforcement learning (Shani et al., 2024; Abdulhai et al., 2023) when aligning Large Language Models (LLMs) with human intent, which is known as Reinforcement Learning from Human Feedback (RLHF) (Bai et al., 2025; Ouyang et al., 2022). Early RLHF pipelines typically involve training a reward model from pairwise comparisons to predict human preferences and guide a policy model through reinforcement learning algorithms like Proximal Policy Optimization (PPO) (Schulman et al., 2017). Despite their effectiveness, PPO introduces intensive computational overhead and is sensitive to reward model inaccuracies, motivating value-free alternatives such as Group Relative Policy Optimization (GRPO) (Shao et al., 2024). Adopted by leading LLMs including OpenAI-o1 (Jaech et al., 2024) and DeepSeek-R1 (Guo et al., 2025), GRPO aims to optimize policies based on relative preferences within a group of samples, providing a robust signal for policy improvement, particularly when absolute rewards are difficult to define or noisy. These advancements in LLM alignment provide a strong foundation for exploring similar human-centric optimization strategies in visual generation domains.

**Alignment for Flow Models**. Diffusion and Flow models (Ho et al., 2020; Song et al., 2020a;b; Peebles & Xie, 2023; Rombach et al., 2022), which offer flexible visual creation through an iterative denoising process, have revolutionized the field of visual synthesis and become a pivotal part of generative models. Building on the success of aligning LLMs with human preferences, similar techniques have recently been transplanted to diffusion and flow models (Podell et al., 2023; Esser et al., 2024; Labs, 2024). Pioneer works like DDPO (Black et al., 2023) and ReFL (Xu et al., 2023) apply PPO to finetune diffusion models for improved aesthetic performance and human feedback alignment. These methods face challenges inherent to RL, including high variance, low efficiency, and sparse reward. Diffusion-DPO (Wallace et al., 2024) adapts the Direct Preference Optimization (DPO) (Rafailov et al., 2023) framework to directly optimize diffusion models from paired preference data, bypassing the need for an explicit reward model but suffering from distribution shift since no new samples are collected during training. Recent efforts such as DanceGRPO (Xue et al., 2025) and Flow-GRPO (Liu et al., 2025) enable GRPO-style policy updates by converting the ODE sampling into an equivalent SDE to each timestep, thereby acquiring a group of denoising directions for statistical sampling and RL exploration for flow models. More recently, Mix-GRPO (Li et al., 2025) has improved training efficiency through a hybrid ODE-SDE sampling approach while maintaining comparable performance. However, these methods are generally constrained by sparse rewards due to long-range stochasticity injection and the binding of each sampling direction to a fixed denoising granularity. These paradigms restrict the ability to conduct a comprehensive evaluation of each sampling direction, limiting the optimization ceiling of GRPO training.

## 3 PRELIMINARY

For the flow-based GRPO methods (Xue et al., 2025; Liu et al., 2025; Li et al., 2025), the denoising process is first modeled as a multi-step Markov decision process (MDP). Given a prompt $\boldsymbol{c}$, the agent with a flow model $p_\theta$ produce a reverse-time trajectories defined as $\Gamma = (\mathbf{s}_T, \mathbf{a}_T, \mathbf{s}_{T-1}, \mathbf{a}_{T-1}, \ldots, \mathbf{s}_0, \mathbf{a}_0)$, where $\mathbf{s}_t = (\boldsymbol{c}, t, \boldsymbol{x}_t)$ is the state at timestep $t$ and $\boldsymbol{x}_t$ is the corresponding noisy sample. Specifically, $\boldsymbol{x}_T \sim N(0, I)$ and $\boldsymbol{x}_0$ is the denoised image. The action $\mathbf{a}_t$ represents the single step denoising process with the policy $\pi_\theta$, indicating a sampling direction $\frac{d\boldsymbol{x}}{dt}$ from $\boldsymbol{x}_t$ to $\boldsymbol{x}_{t-1}$, i.e. $\boldsymbol{x}_{t-1} \sim \pi_\theta(\boldsymbol{x}_{t-1}|\boldsymbol{x}_t, \boldsymbol{c})$.

**SDE Sampling.** As an online RL algorithm, GRPO needs grouped outputs and relative advantages to optimize policies. However, the flow matching model utilizes a deterministic ODE to sample the denoising direction:

$$d\boldsymbol{x}_t = \boldsymbol{v}_\theta(\boldsymbol{x}_t, t)dt, \tag{1}$$

where, $\boldsymbol{v}_\theta(\boldsymbol{x}_t, t)$ is the model output, given the noisy sample $\boldsymbol{x}_t$ and timestep $t$.

To match GRPO's stochastic sampling requirements, Flow-GRPO (Liu et al., 2025) converts the ODE into an equivalent SDE sampling with the same marginal distribution:

$$d\boldsymbol{x}_t = \left( \boldsymbol{v}_\theta \left( \boldsymbol{x}_t, t \right) + \frac{\sigma_t^2}{2t} \left( \boldsymbol{x}_t + (1-t) \boldsymbol{v}_\theta \left( \boldsymbol{x}_t, t \right) \right) \right) dt + \sigma_t d\boldsymbol{w}_t, \tag{2}$$

where $d\boldsymbol{w}_t$ denotes Wiener process increments, and $\sigma_t$ controls the stochasticity injected into the sampling direction. Furthermore, it can be discretized via the Euler–Maruyama scheme:

$$\boldsymbol{x}_{t+\Delta t} = \boldsymbol{x}_t + \left( \boldsymbol{v}_\theta \left( \boldsymbol{x}_t, t \right) + \frac{\sigma_t^2}{2t} \left( \boldsymbol{x}_t + (1-t) \boldsymbol{v}_\theta \left( \boldsymbol{x}_t, t \right) \right) \right) \Delta t + \sigma_t \sqrt{\Delta t} \epsilon, \quad \epsilon \sim \mathcal{N}(0, I). \tag{3}$$

As defined in Flow-GRPO, $\sigma_t = \eta \sqrt{\frac{t}{1-t}}$, where the noise level is controlled by hyperparameter $\eta$.

**GRPO Training.** With SDE sampling, flow-based GRPO methods introduce stochasticity at each timestep to generate a group of $G$ images $\{\boldsymbol{x}_0^i\}_{i=1}^G$. Then the reward model assigns a score $R(\boldsymbol{x}_0^i, \boldsymbol{c})$ to $\boldsymbol{x}_0^i$, and the advantage is computed as:

$$A_t^i = \frac{R(\boldsymbol{x}_0^i, \boldsymbol{c}) - \text{mean}(\{R(\boldsymbol{x}_0^j, \boldsymbol{c})\}_{j=1}^G)}{\text{std}(\{R(\boldsymbol{x}_0^j, \boldsymbol{c})\}_{j=1}^G)}. \tag{4}$$

Note that the advantages $A_0^i$ obtained from the final step image are uniformly broadcast to each step $A_t^i$ to evaluate the SDE sampling directions. Finally, the policy model is optimized by maximizing the following objective:

$$\mathcal{J}_{\text{Flow-GRPO}}(\theta) = \mathbb{E}_{\boldsymbol{c} \sim \mathcal{C}, \{\boldsymbol{x}^i\}_{i=1}^G \sim \pi_{\theta_{\text{old}}}(\cdot|\boldsymbol{c})} f(r, A, \theta, \varepsilon, \beta), \tag{5}$$

where

$$f(r, A, \theta, \varepsilon, \beta) = \frac{1}{G} \sum_{i=1}^G \frac{1}{T} \sum_{t=0}^{T-1} \left( \min \left( r_t^i(\theta) A_t^i, \text{clip} \left( r_t^i(\theta), 1 - \varepsilon, 1 + \varepsilon \right) A_t^i \right) - \beta D_{\text{KL}} \left( \pi_\theta \| \pi_{\text{ref}} \right) \right), \tag{6}$$

$$r_t^i(\theta) = \frac{p_\theta \left( \boldsymbol{x}_{t-1}^i \mid \boldsymbol{x}_t^i, \boldsymbol{c} \right)}{p_{\theta_{\text{old}}} \left( \boldsymbol{x}_{t-1}^i \mid \boldsymbol{x}_t^i, \boldsymbol{c} \right)}. \tag{7}$$

Notably, $\beta$ is the hyperparameter that controls the proportion of KL loss. Following the practices of DanceGRPO and MixGRPO, we set $\beta = 0$ to achieve a more stable training process.

## 4 GRANULAR-GRPO

As an online RL algorithm, flow-based GRPO methods utilize SDE to sample a group of denoising directions for optimization. A core issue underlying this paradigm is to obtain precise and comprehensive assessments of each sampling direction. To this end, we introduce the $\text{G}^2\text{RPO}$ framework to (i) confine stochasticity to individual steps (Singular Stochastic Sampling) for more precise reward signals, and (ii) integrate the advantages derived from multi-granularity denoising results (Multi-Granularity Advantage Integration) to acquire a more comprehensive evaluation, as shown in Fig. 2.

### 4.1 SINGULAR STOCHASTIC SAMPLING

Traditional flow-based GRPO methods introduce SDE sampling at every step to inject stochasticity and uniformly assign the final image reward to each step's sampling direction. According to Eq. 4, the advantage $A_t^i$ of each sampling direction at step $t$ is equally assigned with $A_0^i$. However, the reward signal available only after multiple decision steps impedes the model's capability to link the final reward to each decision, thereby resulting in imprecise and sparse rewards.

To acquire a dense reward for each SDE sampling direction, a simple yet effective strategy is to confine the stochasticity to the single step selected for optimization. Firstly, we designate a set of candidate SDE timesteps $M \subset \{1, \ldots, T\}$ with $|M| = K \leq T$. As shown in Fig. 2, given a prompt $\boldsymbol{c}$ and initial noise $\boldsymbol{x}_T$, each timestep denoted by $k \in M$ will be optimized in the training phase. Then, a common starting point $\boldsymbol{x}_k$ for the group is acquired using ODE sampling from Eq. 1.

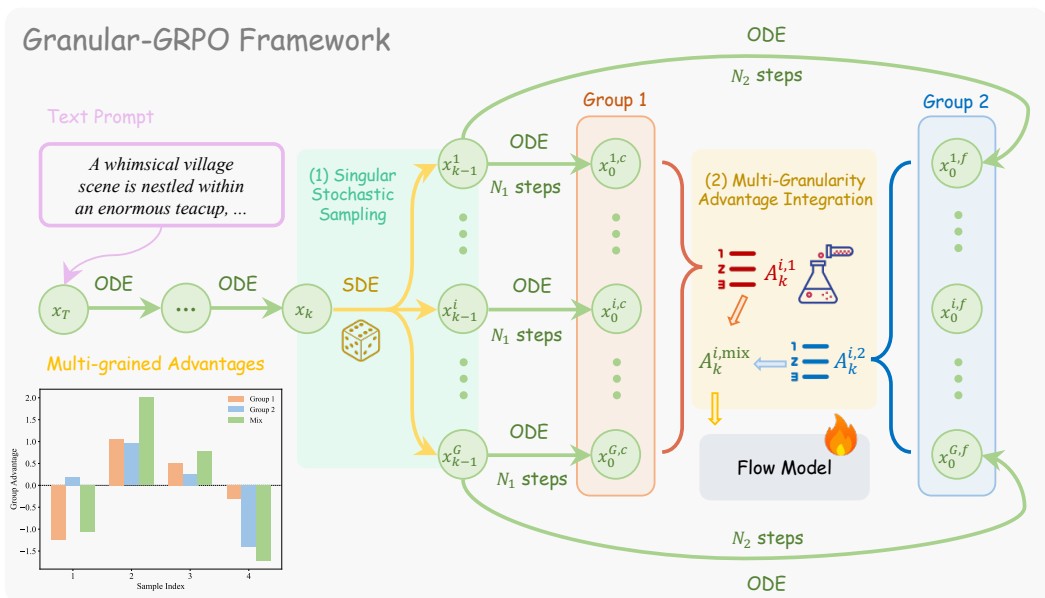

Figure 2: **Overview of $G^2RPO$** . Given a text prompt and an initial noise, our Singular Stochastic Sampling strategy employs SDE sampling solely at a single step and samples a group of distinct denoising directions. Then, the Multi-Granularity Advantage Integration module executes multi-granularity ODE denoising for each direction and integrates the advantages to produce a comprehensive evaluation for each sampling direction. For simplicity, the figure shows one coarse-grained path (denoted as $c$) and one fine-grained path (denoted as $f$).

Notably, the Singular Stochastic Sampling strategy employs SDE sampling only at $\boldsymbol{x}_k$ and samples $G$ distinct denoising directions to get the next noisy state $\{\boldsymbol{x}_{k-1}^i\}_{i=1}^G$. Then each $\boldsymbol{x}_{k-1}^i$ undergoes $k-1$ steps of ODE sampling to generate a deterministic denoised image $\boldsymbol{x}_{0\leftarrow k}^i$. Based on our sampling strategy, the variance of the group reward $\{R(\boldsymbol{x}_{0\leftarrow k}^i, \boldsymbol{c})\}_{i=1}^G$ is entirely determined by the distinct denoising directions introduced by the SDE sampling at step $k$. And a step-aware, precise advantage can be acquired:

$$A_k^i = \frac{R(\boldsymbol{x}_{0\leftarrow k}^i, \boldsymbol{c}) - \text{mean}(\{R(\boldsymbol{x}_{0\leftarrow k}^i, \boldsymbol{c})\}_{i=1}^G)}{\text{std}(\{R(\boldsymbol{x}_{0\leftarrow k}^i, \boldsymbol{c})\}_{i=1}^G)}. \tag{8}$$

Consequently, the $f(r, A, \theta, \varepsilon, \beta)$ in Eq 5 can be formulated as:

$$f(r, A, \theta, \varepsilon, \beta) = \frac{1}{G} \sum_{i=1}^G \frac{1}{K} \sum_{k \in M} \left( \min \left( r_k^i(\theta) A_k^i, \text{clip} \left( r_k^i(\theta), 1 - \varepsilon, 1 + \varepsilon \right) A_k^i \right) \right). \tag{9}$$

After the sampling phase is completed, the training of GRPO requires the computation of $p_\theta \left( \boldsymbol{x}_{k-1}^i \mid \boldsymbol{x}_k^i, \boldsymbol{c} \right)$ to obtain $r_k^i(\theta)$ refer to eq 7. In practice, each distinct sampling starting point $\boldsymbol{x}_k^i$ needs to be fed into the flow model to compute the corresponding ODE denoising direction $\boldsymbol{v}_k^i$. However, our sampling strategy shares a common starting point $\boldsymbol{x}_k$, allowing a group of $G$ samples to reuse the same $\boldsymbol{v}_k$, which in turn improves training efficiency.

## 4.2 MULTI-GRANULARITY ADVANTAGE INTEGRATION

Singular Stochastic Sampling accurately constrains stochasticity into the single SDE step, ensuring a strong correlation between the reward and the injected noise. Nevertheless, how to acquire a comprehensive reward for the denoising direction of the current step still requires further investigation.

As shown in Fig. 3, we observe that under identical $\boldsymbol{x}_k$ and prompt $\boldsymbol{c}$ conditions, the denoising trajectory generated by singular stochastic sampling is not robust when assessing the corresponding SDE denoising direction. Images generated from denoising trajectories with different granularities show similar overall content but exhibit discrepancies in detail due to the varying denoising intervals.

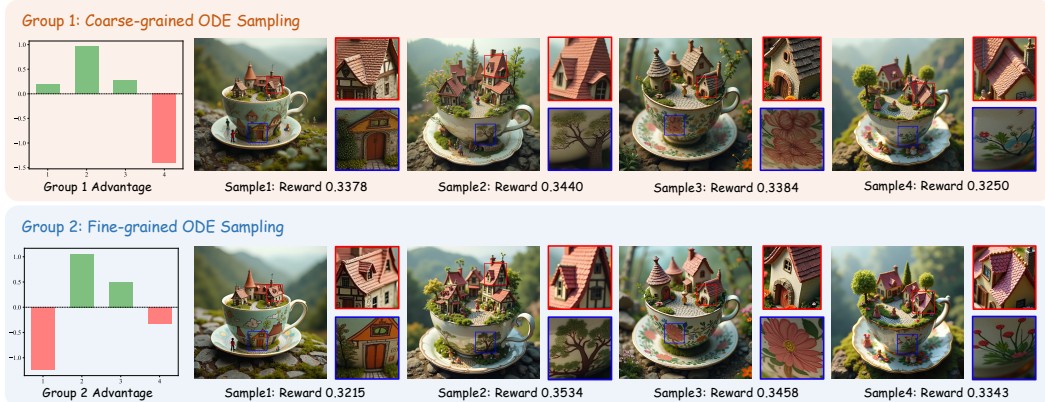

Figure 3: **Visual Comparison of Images Denoised at Different Granularities.** Images denoised at different granularities exhibit variations in fine details and textures, leading to inconsistent scoring by the Reward Model (HPS-v2.1). This observation reveals the insufficiency of a single-granularity evaluation of group advantage.

Such differences are evident in the scores assigned by the Reward Model, further influencing the numerical values of the advantage within the group, and even the optimization direction.

To this end, we propose a Multi-Granularity Advantage Integration module to perform multi-granularity denoising on the sampled denoising directions within a group. The advantages of the images denoised at different granularities are then integrated to form the final evaluation. Specifically, as shown in Fig. 2, step $k$ is the SDE sampling step, and $G$ distinct denoising directions are sampled to acquire next noisy state $\{x_{k-1}^i\}_{i=1}^G$. Under the conventional granularity condition, each $x_{k-1}^i$ undergoes $k-1$ steps to obtain the final denoised image. The sequence of denoising timesteps can be represented as: $\mathcal{S} = \{1, 2, \ldots, k-1\}$. For our Multi-Granularity denoising module, a set of integer scaling factors $\Lambda = \{\lambda_1, \lambda_2, ..., \lambda_j\}, |\Lambda| = J$ is defined to represent different denoising granularities. Each $\lambda_j$ implements interval sampling for different denoising granularity, that means sample every $\lambda_j$-th step from the total $k-1$ steps. The denoising timestep sequence $\mathcal{S}_j$ can be formally represented as:

$$\mathcal{S}_j = \{1, 1 + \lambda_j, 1 + 2\lambda_j, \ldots, \left\lceil \frac{k-1}{\lambda_j} \right\rceil \lambda_j\}, \tag{10}$$

Our interval sampling approach ensures that the denoising process is performed at regular intervals defined by $\lambda_j$. As $\lambda_j$ increases, the granularity becomes coarser, allowing for a more flexible and adaptive denoising process. For ease of illustration, $J = 2$ in the Fig. 2.

After $N_j$ subsequent steps denoising for $\{x_{k-1}^i\}_{i=1}^G$, a group of noise-free images $\{x_0^{i,j}\}_{i=1}^G$ are generated. Subsequently, different groups images gets the reward $\{R(x_{0 \leftarrow k}^{i,j}, c)\}_{i=1}^G$ from a reward model and then compute the intra-group advantages $\{A_t^{i,j}\}_{i=1}^G$ with Eq. 4. Similar to the joint training with multiple reward models (e.g., HPS-v2.1 and CLIP Score) in DanceGRPO, where the advantages from different reward models are directly summed, we combines the advantages from different granularities to get $\{A_t^{i,\mathrm{mix}}\}_{i=1}^G$:

$$A_t^{i,\mathrm{mix}} = \sum_j^J A_t^{i,j} \tag{11}$$

Finally, $f(r, A, \theta, \varepsilon, \beta)$ is updated to:

$$f(r, A, \theta, \varepsilon, \beta) = \frac{1}{G} \sum_{i=1}^G \frac{1}{K} \sum_{k \in M} \left( \min \left( r_k^i(\theta) A_k^{i,\mathrm{mix}}, \mathrm{clip} \left( r_k^i(\theta), 1 - \varepsilon, 1 + \varepsilon \right) A_k^{i,\mathrm{mix}} \right) \right), \tag{12}$$

and using Eq. 5 to optimize the policy $\pi_\theta$. A detailed algorithm is illustrated in Algorithm 1.

---

**Algorithm 1** G$^2$RPO Training Process

---

1: **Require:** Prompt dataset $\mathcal{C}$, policy model $\pi_\theta$, reward model $R$, total sampling steps $T$
2: **Require:** SDE sampling timestep set $M$, Denoising granularities set $\Lambda$ ($|\Lambda| = J$)
3: **for** training iteration $e = 1$ **to** $E$ **do**
4:     Update old policy model: $\pi_{\theta_{old}} \leftarrow \pi_\theta$
5:     Sample batch prompts $\mathcal{C}_b \sim \mathcal{C}$
6:     **for** prompt $\mathbf{c} \in \mathcal{C}_b$ **do**
7:         Init same noise $\boldsymbol{x}_T \sim \mathcal{N}(0, \mathbf{I})$
8:         **for** $k \in M$ **do**
9:             **for** $t = T$ to $0$ **do**
10:                 **if** $t > k$ **then**
11:                     ODE Sampling: $\boldsymbol{x}_{t-1}$
12:                 **else if** $t == k$ **then**
13:                     SDE Sampling a group samples: $\boldsymbol{x}_{k-1}^i$     // $i$-th direction in the group
14:                 **else if** $t > k$ **then**
15:                     **for** $\lambda_j \in \Lambda$ **do**
16:                         ODE Sampling with granularity $\lambda_j$: $\boldsymbol{x}_{t-1}^{i,j}$
17:                     **end for**
18:                 **end if**
19:             **end for**
20:         Get a group of reward: $R(\boldsymbol{x}_{0 \leftarrow k}^{i,j})$
21:         $A_k^j \leftarrow \frac{R(\boldsymbol{x}_{0 \leftarrow k}^{i,j}, \boldsymbol{c}) - \mu^j}{\sigma^j}$
22:         $A_k^{mix} \leftarrow \sum_{j=1}^J A_k^j$
23:         **end for**
24:     Compute GRPO loss $J(\theta)$
25:     **end for**
26:     Update policy: gradient ascent on $J(\theta)$
27: **end for**

---

## 5 EXPERIMENTS

### 5.1 IMPLEMENTATION DETAILS

**Datasets and Backbone.** Following DanceGRPO (Xue et al., 2025) and MixGRPO (Li et al., 2025), we evaluate our G$^2$RPO using the HPSv2 (Wu et al., 2023) dataset. It contains 103,700 text prompts for training and 400 diverse prompts for testing. The text-to-image model employed for reinforcement learning is Flux.1-dev (Labs, 2024), a leading flow model in the community.

**Evaluation Metrics.** To evaluate the effectiveness and robustness of our G$^2$RPO , multiple reward models are applied as evaluation metrics. These reward models assess the alignment between generated images and human preferences from multiple dimensions. Specifically, HPS-v2.1 (Wu et al., 2023), CLIP Score (Radford et al., 2021), and Pick Score (Kirstain et al., 2023) collectively assess semantic alignment and visual coherence. Image Reward (Xu et al., 2023) focuses on visual quality and aesthetic appeal, while Unified Reward (Wang et al., 2025) is the SOTA unified reward model that comprehensively evaluates both alignment with the caption and overall image quality.

**Evaluation Setting.** Similar to DanceGRPO and MixGRPO, two experimental settings are employed. Firstly, a single HPS-v2.1 reward model is utilized for training to verify the upper limit of improvement for in-domain performance. However, as demonstrated by DanceGRPO, HPS-v2.1 is prone to model hacking due to biases in the training set, leading to degradation in other evaluation metrics. Therefore, our primary experiment also involves joint training with both HPS-v2.1 and CLIP Score as reward models to acquire stable and robust results.

**Sampling Phase.** Following DanceGRPO, a shared initialization noise is used to generate a group of 12 images from the same text prompt. The total sampling step $T = 16$ to enhance computational efficiency and the advantage clip $\varepsilon = 5$ in Eq. 6. Note that, the parameter $\eta$ in Eq. 3 directly determines the noise level, which in turn defines the size of the stochastic exploration space in the

SDE. Leveraging the Singular Stochastic Sampling strategy, our precise reward can tolerate a larger $\eta = 0.7$. The set of candidate SDE timesteps $M$ consists of the first 8 timesteps to improve training efficiency. For the Multi-Granularity Advantage Integration strategy, the set of distinct granularities $\Lambda = \{1, 2, 3\}$.

**Training Phase.** All experiments are conducted using $16\times$ NVIDIA H200 GPUs, with a batch size of 1. The AdamW optimizer is used, configured with a learning rate of $2 \times 10^{-6}$ and a weight decay of $1 \times 10^{-4}$. Mixed precision training is implemented using bfloat16 (bf16) format. The training iteration is 300. More detailed parameters refer to the Appendix Section B.

## 5.2 Main Results

**Quantitative Evaluation.** As shown in Tab. 1, DanceGRPO and MixGRPO serve as the baselines for comparison with our $G^2$RPO . Additionally, the performance of employing the Singular Stochastic Sampling strategy without multi-granularity denoising ($G^2$RPO w/o MGAI) is also proposed. It can be observed that when HPS-v2.1 is used solely as the training reward model, our Singular Stochastic Sampling achieves a relative improvement of 6.52% compared to DanceGRPO. This indicates that constraining the stochasticity to a single step yields a precise reward signal, which in turn provides a more faithful optimization signal and enables GRPO to enhance the optimization ceiling. However, as demonstrated by DanceGRPO, optimizing solely with HPS-v2.1 can induce model hacking, which in turn compromises other out-of-domain evaluation metrics. Secondly, for the setting of multi-reward (HPS-v2.1 and CLIP Score) optimization, the results indicate that $G^2$RPO attains superior performance across both in-domain and out-of-domain rewards. Under the multi-granularity denoising condition, groups of images at different granularities provide a more comprehensive evaluation for the SDE sampling direction. This multi-granularity paradigm also allows for more flexible adaptation to the preferences of various reward models, thereby achieving significant improvements in various out-of-domain dimensions. Additional experiment settings and results refer to the Appendix (Section C).

Table 1: **Quantitative Results.** Comparison of results on in-domain and out-of-domain rewards.

| Reward Model | Method | In-Domain | | Out-of-Domain | | |
| --- | --- | --- | --- | --- | --- | --- |
| | | HPS-v2.1 | CLIP Score | Pick Score | ImageReward | Unified Reward |
| / | Flux.1-dev | 0.305 | 0.388 | 0.226 | 1.040 | 3.621 |
| HPS-v2.1 | DanceGRPO | 0.353 | **0.375** | 0.228 | 1.233 | **3.548** |
| | MixGRPO | 0.378 | 0.358 | 0.225 | 1.266 | 3.421 |
| | $G^2$RPO w/o MGAI | 0.376 | 0.351 | 0.228 | 1.286 | 3.469 |
| | **$G^2$RPO** | **0.385** | 0.355 | **0.229** | **1.313** | 3.487 |
| HPS-v2.1 & CLIP | DanceGRPO | 0.331 | 0.389 | 0.227 | 1.128 | 3.569 |
| | MixGRPO | 0.363 | 0.399 | 0.230 | 1.436 | 3.661 |
| | $G^2$RPO w/o MGAI | 0.372 | 0.395 | 0.234 | 1.421 | 3.688 |
| | **$G^2$RPO** | **0.376** | **0.406** | **0.235** | **1.483** | **3.783** |

**Qualitative Comparison.** Fig. 4 presents the qualitative comparison among the original Flux.1-dev, DanceGRPO, MixGRPO, and our proposed $G^2$RPO . It can be observed that $G^2$RPO provides enhanced detail fidelity and improves the consistency with the text prompt, achieving superior alignment with human preferences. For instance, in the second column, our $G^2$RPO faithfully captures the specified expressions and even the nuances of the chess pieces as described in the prompt, delivering finer details and higher visual quality. Moreover, in the "poster" case depicted in the last column, $G^2$RPO not only adheres to the spatial requirement of a clear left-right demarcation but also renders the reflections of the trees with remarkable clarity. Additionally, the overall style of the image generated by $G^2$RPO is more consistent with the aesthetic demands of poster design.

## 5.3 Ablation Study

As described in Section 4, the Multi-Granularity Advantage Integration module integrates multi-granularity ODE sampling results to evaluate the SDE sampling direction comprehensively. Different granularities represent diverse sampling intervals during denoising, controlled by the parameter set $\Lambda$. To validate the effectiveness of multi-granularity fusion, we perform ablation studies on various $\Lambda$ set shown in Tab. 2. It can be found that as the number of selected granularities increases,

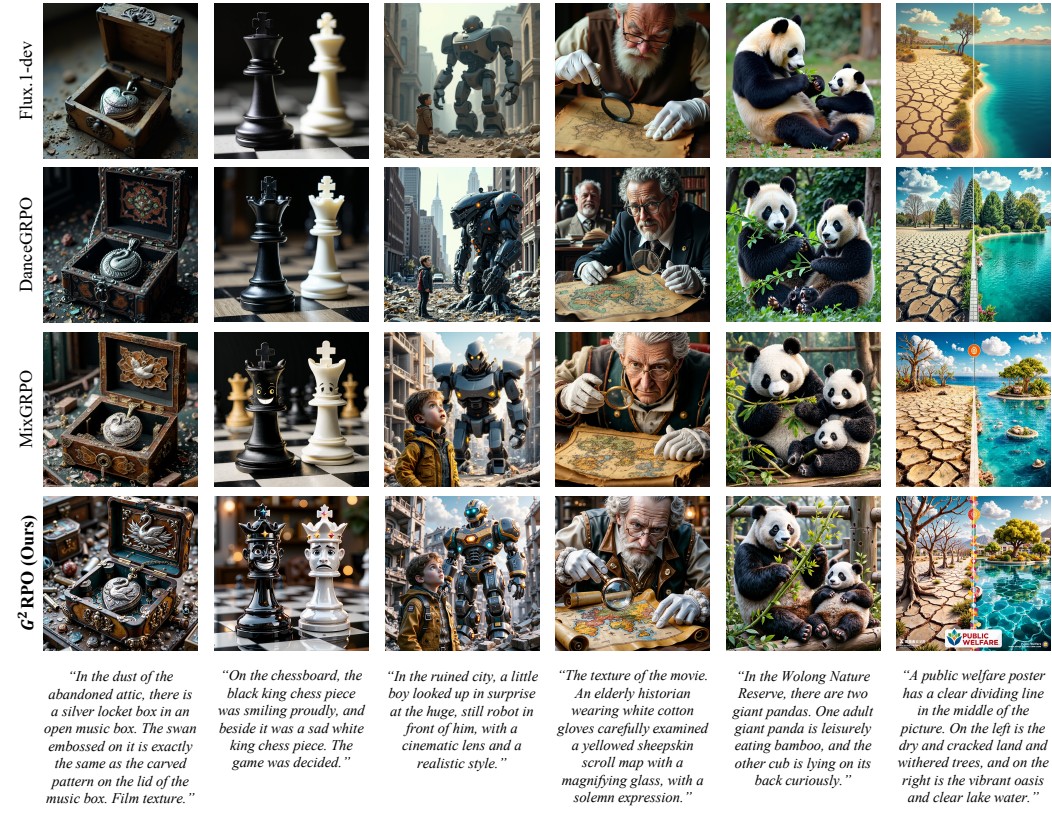

Figure 4: **Qualitative Results.** Comparison with existing flow-based GRPO methods, in which our $G^2$RPO demonstrates superior performance in human preference alignment.

the evaluation of each sampling direction becomes more comprehensive, facilitating a robust assessment through multi-granularity advantage fusion, which significantly improves performance across both in-domain and out-of-domain reward models.

Table 2: **Ablation Study.** Comparison for different denoising granularities.

| Reward Model | $\Lambda$ | In-Domain | | Out-of-Domain | | |
|---|---|---|---|---|---|---|
| | | HPS-v2.1 | CLIP Score | Pick Score | ImageReward | Unified Reward |
| HPS-v2.1 & CLIP | $\{1\}$ | 0.372 | 0.395 | 0.234 | 1.421 | 3.688 |
| | $\{1, 2\}$ | 0.375 | 0.404 | 0.234 | 1.468 | 3.759 |
| | $\{1, 3\}$ | **0.378** | 0.404 | 0.234 | 1.465 | 3.760 |
| | $\{1, 2, 3\}$ | 0.376 | **0.406** | **0.235** | **1.483** | **3.783** |

## 6 CONCLUSION

This paper addresses the critical limitations of precisely evaluating the quality of denoising directions sampled by Flow-based GRPO for human preference alignment. We introduce $G^2$RPO , a novel online RL framework that precisely localizes stochasticity to a single step within the denoising process and provides a comprehensive evaluation of SDE denoising directions by integrating the advantages derived from images at different denoising granularities. This innovative design enables the provision of dense, precise reward signals, thereby fundamentally improving optimization accuracy and leading to a more robust and higher-quality alignment. Our extensive experiments consistently demonstrate that $G^2$RPO achieves superior performance across diverse reward conditions, marking a significant advancement in aligning generative models with human preferences.

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

## A  APPENDIX

In the appendix, we present detailed hyperparameter settings (Section B), further exploration of varying inference steps (Section C), the limitations of our method (Section E), the ethical statement (Section F), the declaration on LLM Usage (Section G), as well as more qualitative evaluation results (Section D).

## B  HYPERPARAMETER SETTINGS

Tab. 3 shows the detailed hyperparameter configuration used in our experiments.

Table 3: Hyperparameter settings used in all experiments.

| Parameter | Value | Parameter | Value |
|---|---|---|---|
| Training: | | | |
| Random seed | 42 | Learning rate | $2 \times 10^{-6}$ |
| Train batch size | 1 | Weight decay | $1 \times 10^{-4}$ |
| Warmup steps | 0 | Mixed precision | bfloat16 |
| Dataloader workers | 4 | Max grad norm | 1.0 |
| Resolution | $720 \times 720$ | Sampling steps | 16 |
| Eta | 0.7 | Sampler seed | 1223627 |
| Group size | 12 | Scheduler shift | 3 |
| Clip range | $1 \times 10^{-4}$ | Adv. clip max | 5.0 |
| Init same noise | Yes | SDE steps $M$ | 16 15 14 13 12 11 10 9 |
| Denoising granularity $\Lambda$ | $\{1, 2, 3\}$ | | |
| Inference: | | | |
| Resolution | $1024 \times 1024$ | Sampling steps | 50 |

## C  FURTHER EXPLORATION OF VARYING INFERENCE STEPS

In the main experiment Tab. 1, we maintained the same inference settings as DanceGRPO and MixGRPO, generating images with a resolution of $1024 \times 1024$ in 50 steps. Notably, our MGAI module, which evaluates denoising samples of different granularities in a mixed manner, provides a more comprehensive assessment. With the assistance of this module, G$^2$RPO exhibits stronger robustness to varying denoising step configurations. As shown in Tab. 4, all flow-based GRPO methods are jointly trained with HPS-v2.1 and CLIP. When the total inference timesteps are reduced to 20 or even 10 steps, G$^2$RPO still achieved significant performance improvements across various in-domain and out-of-domain evaluation reward models.

Table 4: Comprehensive evaluation of total denoising steps.

| Reward Model | Method | In-Domain | | Out-of-Domain | | |
|---|---|---|---|---|---|---|
| | | HPS-v2.1 | CLIP Score | Pick Score | ImageReward | Unified Reward |
| 10 Step Inference | Flux.1-dev | 0.289 | 0.388 | 0.225 | 0.939 | 3.504 |
| | DanceGRPO | 0.325 | 0.390 | 0.227 | 1.129 | 3.576 |
| | MixGRPO | 0.358 | 0.401 | 0.230 | 1.431 | 3.641 |
| | **G$^2$RPO** | **0.378** | **0.408** | **0.235** | **1.519** | **3.805** |
| 20 Step Inference | Flux.1-dev | 0.300 | 0.389 | 0.226 | 1.034 | 3.575 |
| | DanceGRPO | 0.329 | 0.388 | 0.228 | 1.136 | 3.586 |
| | MixGRPO | 0.363 | 0.401 | 0.230 | 1.430 | 3.651 |
| | **G$^2$RPO** | **0.376** | **0.407** | **0.235** | **1.511** | **3.806** |

## D  ADDITIONAL QUALITATIVE EVALUATION

We provide additional qualitative evaluation results shown in Fig. 5 and Fig. 6.

## E  LIMITATION AND FUTURE WORKS

Despite the advancements of our $G^2RPO$ in human preference alignment with flow-based GRPO training, it faces certain constraints. Specifically, $G^2RPO$ incurs additional sampling time due to multi-granularity sampling, particularly when a larger number of granularities are selected. However, it is important to note that, as illustrated in Fig. 1 (a), with the aid of precise and comprehensive rewards, our $G^2RPO$ achieves the upper limit of DanceGRPO within only one-fourth of the iteration rounds. In future work, we will apply additional reward models (such as PickScore and Unified Reward) for GRPO training to explore the preferences of different reward models. Meanwhile, we will also explore the application of $G^2RPO$ to more generation tasks, such as text-to-video generation and image-to-video generation, etc.

## F  ETHICAL STATEMENT

In this research, we reaffirm our dedication to maintaining the highest ethical standards and fostering responsible innovation. We are aware that the outputs generated by GRPO may be influenced by the biases inherent in the reward models used. However, upon thorough examination, we have not identified any content that violates ethical norms or guidelines. Our study does not involve any data, methodologies, or applications that pose ethical concerns. All experiments and analyses were conducted in strict adherence to established ethical protocols, ensuring the integrity and transparency of our research.

## G  DECLARATION ON LLM USAGE

In this work, LLM is utilized only for minor language refinement.

## H  REPRODUCIBILITY STATEMENT

In an effort to ensure the full reproducibility of our research and to contribute to the broader academic community, we will publicly release the model checkpoint and the complete source code for both training and inference. We anticipate that these resources will serve as a valuable reference for future flow-based GRPO research, thereby fostering innovation and accelerating progress within the community.

| Flux.1-dev | DanceGRPO | MixGRPO | $G^2$ RPO (Ours) |

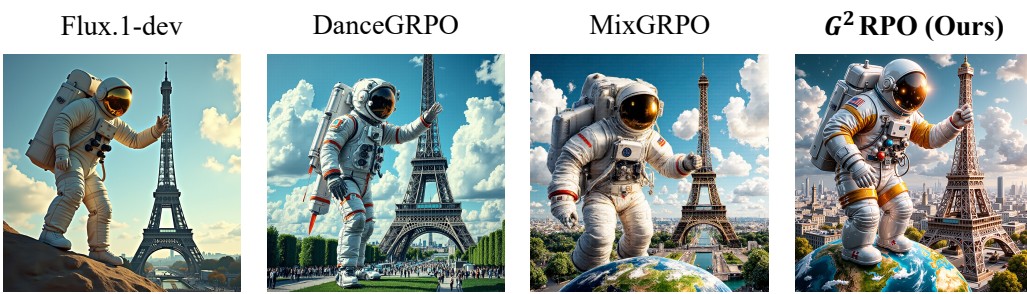

*"Please generate a picture: an astronaut as huge as a mountain, landing on earth, curiously touching the spire of the Eiffel Tower in Paris."*

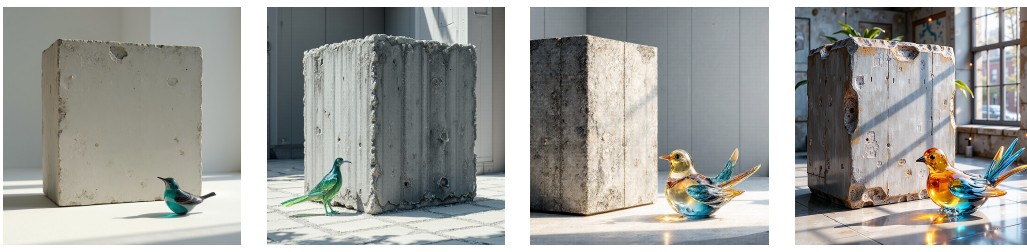

*"Next to a huge rough concrete square, there is a small and exquisite glass bird. The picture adopts a minimalist style with clear light and shadow."*

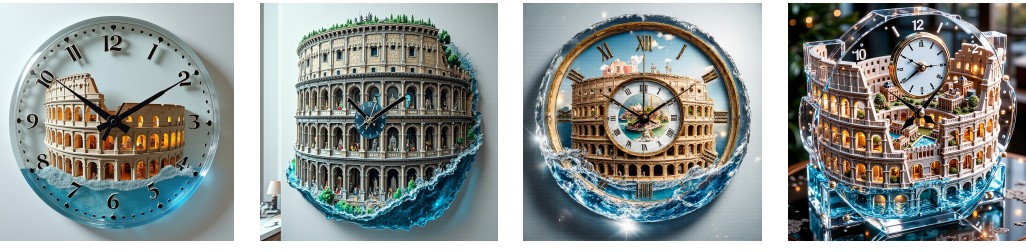

*"A crystal wall clock in the shape of an ancient Roman Colosseum, inside the clock is a miniature city."*

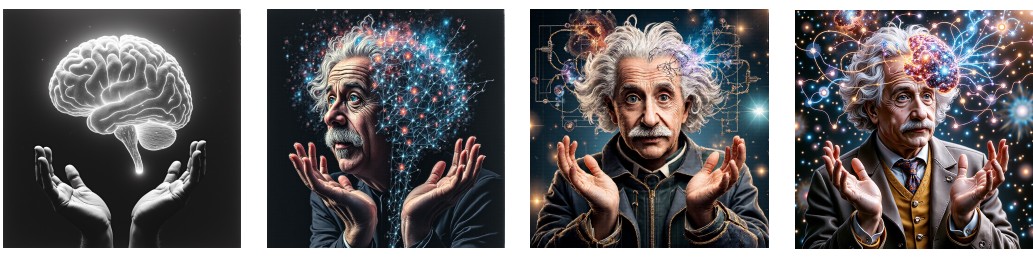

*"Albert Einstein used his hands to create a brain-shaped nebula, whose lines resembled a complex electrical diagram."*

Figure 5: **Qualitative comparison with existing GRPO methods. Best viewed zoomed in.**

| Flux.1-dev | DanceGRPO | MixGRPO | $G^2$ RPO (Ours) |
|---|---|---|---|

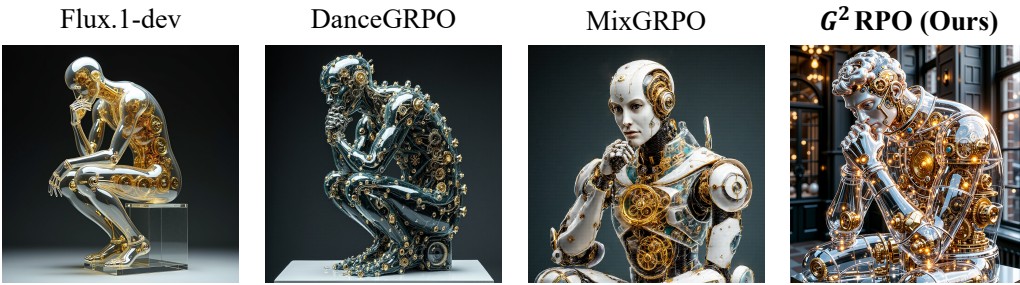

*"Please create a sculpture. The main body is a robot that imitates Rodin's "The Thinker". The whole body is made of transparent glass and has complex golden gears running inside, in a steampunk style.."*

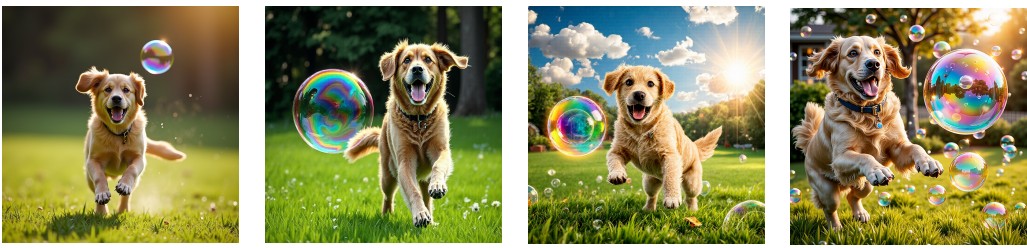

*"A golden Labrador retriever is leaping excitedly on the green grass, chasing a soap bubble that glows with a rainbow in the sun, National Geographic photography style."*

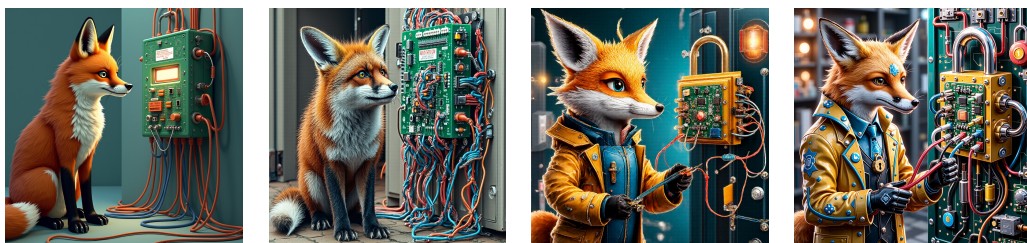

*"A biochemically modified fox faced a complex electronic lock. Instead of forcibly destroying it, it observed the wires and unplugged one of the key wires."*

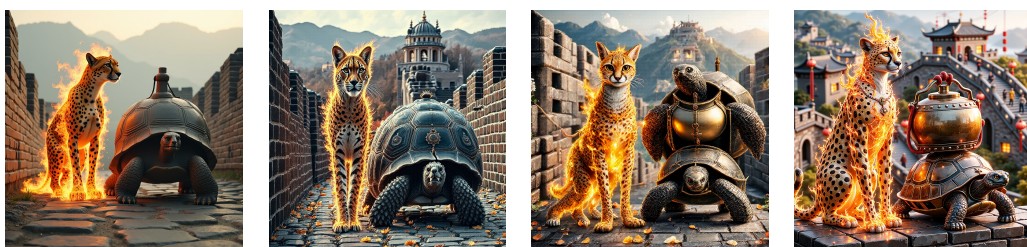

*"Please generate a picture: On the Great Wall, a cheetah with a body burning like a flame is standing side by side with a turtle carrying a huge heavy bronze bell, in sharp contrast."*

Figure 6: **Qualitative comparison with existing GRPO methods. Best viewed zoomed in.**

