# OpenReview forum: "$\text{G}^2$RPO: Granular GRPO for Precise Reward in Flow Models"
_ICLR.cc/2026/Conference — ICLR 2026 Conference Withdrawn Submission_

### Official Review · Reviewer_6ymD · 2025-10-27

**Soundness:** 3
**Presentation:** 2
**Contribution:** 3
**Rating:** 6
**Confidence:** 4

**Summary:**

This paper proposes **G²RPO (Granular-GRPO)**, an online reinforcement learning framework designed to improve reward precision and alignment fidelity for flow-based generative models.
The authors identify two key weaknesses in existing Flow-GRPO and Dance-GRPO methods:
1. **Sparse rewards** resulting from uniformly distributed credit assignment across SDE steps; and
2. **Single-granularity evaluation**, where each sampling direction is tied to a fixed number of denoising steps.

####
To address these, the authors introduce two major innovations:
**Singular Stochastic Sampling :**
confines stochasticity to a single timestep, establishing a strong correlation between injected noise and its reward signal.
**Multi-Granularity Advantage Integration (MGAI):**
aggregates advantage estimates from multiple denoising granularities to obtain a comprehensive evaluation of sampling directions.
Experiments on the Flux.1-dev flow model demonstrate consistent gains across multiple human preference reward models (HPS-v2.1, CLIP Score, PickScore, ImageReward, Unified Reward).

**Strengths:**

**Innovative Reward Structuring:** The combination of Singular Stochastic Sampling and Multi-Granularity Integration effectively addresses reward sparsity and limited evaluation coverage, both being crucial bottlenecks in RL-based alignment of flow models.

**Methodologically Coherent:** The algorithm is well-motivated, mathematically consistent (Eqs. 8–12), and the training process (Algorithm 1) is clearly documented.

**Empirically Strong:** Extensive quantitative and qualitative results (Tabs. 1–2, Figs. 4–6) demonstrate superior alignment with human preferences and improved visual fidelity.

**Reproducibility:** The paper provides full implementation details, hyperparameters, and training setup, aligning with ICLR reproducibility guidelines.

**Clarity and Presentation:** The manuscript is well-written, structured, and easy to follow, with clear figure illustrations and balanced technical depth.

**Weaknesses:**

### (a) Method generality not fully validated.
The chosen granularity set Λ = {1,2,3} appears taskspecific, without analysis of its optimality or adaptability across different architectures.

**Suggestion:** Include cross-task comparisons with varied granularity configurations (e.g., Λ ={1,4}, Λ = {2,3,4}) to clarify the general applicability and sensitivity of Λ.

### (b) Incomplete methodological justification.
The choice of candidate SDE step set M = {16,15,...,9} lacks motivation—why are later timesteps excluded? Similarly, MGAI’s advantage fusion (Eq. 11) adopts an unweighted summation without comparing against alternative strategies (e.g., weighted averaging or attention-based fusion).

**Suggestion:** Provide ablations for different SDE step ranges (e.g., early vs. middle timesteps) and include comparisons among fusion mechanisms to validate the rationale for the current design.

### (c) Missing efficiency and sensitivity analysis.
While the paper states that G²RPO converges faster (“one-fourth of iterations”), it omits quantitative comparisons on computational cost (training time, inference latency)； The sensitivity of the stochastic noise parameter η= 0.7 is also unexplored.

**Suggestion:** Add explicit measurements of training time per iteration and per-sample generation latency under identical GPU settings, and conduct sensitivity analysis for different η values (e.g.,0.5, 0.7, 0.9).

**Questions:**

a. How sensitive is performance to the choice of the SDE step subset M? Would using multiple stochastic steps harm or help training stability?

b. What is the computational overhead (in time or memory) introduced by the MGAI module?

c. Could G²RPO be extended to diffusionbased models or text-to-video generation with minor modifications?

d. How does the method perform under real human preference models rather than automatic reward surrogates?

e. Why is there no direct, controlled comparison between G²RPO and Flow-GRPO under identical experimental conditions (e.g., the same HPSv2 dataset, Flux.1-dev backbone, and reward models)? Such an evaluation would more clearly highlight G²RPO’s improvements over its most relevant baseline.

---

### Official Review · Reviewer_gZPr · 2025-10-31

**Soundness:** 3
**Presentation:** 3
**Contribution:** 3
**Rating:** 6
**Confidence:** 1

**Summary:**

Authors introduce $G^2$RPO, a method that attributes the reward of generated samples to a single stochastic step made during the generation. Moving away from the original Flow-GRPO method which turns the whole flow ODE into a SDE, $G^2$RPO keeps ODE sampling everywhere but a single step which allows the policy to pinpoint exactly where the stochasticity comes from, allowing for more fine-grained "credit assignment". Moreover, after the SDE step, they denoise the groups with different levels of denoising granularities which gives further signal to the policy.

Quantitatively, their approach yields competitive in-domain and out-of-domain scores against the baselines. Qualitatively, their samples adhere very well to the prompt in addition to being visually more appealing.

**Strengths:**

The method is sound and makes sense and the results strongly motivate the approach.

**Weaknesses:**

- See my question about Eq. 9 down below but theoretical justification for it is needed.

**Questions:**

- I am not very familiar with the ODE to SDE formulation but given that you intertwine the two during generation, does that still guarantee that you're learning and sampling from the desired data distribution? It would be good to clarify that in the paper.
- I might be missing something about the justification for Eq. 9 but why is the sum only conducted on the subset of the SDE timesteps? Disregarding the MULTI-GRANULARITY ADVANTAGE INTEGRATION, my understanding is that you go through all SDE timesteps in $k\in M$ and for each $k$ you unroll your trajectories using ODE sampling until you get to the stochastic step $k$ after which you sample $G$ trajectories that get completed with ODE again. So you end up with $G\times M$ trajectories, in that case in Eq. 9, there should be another sum over all the timesteps as well.
- typo in line 14 in Algorithm 1, I think it should be $t<k$.
- MGAI is ablated but I would also like to see the effect of $M$ on the performance.

---

### Official Review · Reviewer_UsSM · 2025-10-31

**Soundness:** 3
**Presentation:** 3
**Contribution:** 2
**Rating:** 2
**Confidence:** 4

**Summary:**

This paper proposes Granular-GRPO, a framework to fine-tune flow models with reinforcement learning. It introduces two techniques to improve training: 1. it introduces one stochastic sampling step to the flow model inference; 2. it chooses different discretization step sizes when collecting data. It conducts empirical studies to demonstrate the performance.

**Strengths:**

1. the paper is overall well-written with illustrative figures;
2. the experiments demonstrate both in-domain and out-of-domain improvement

**Weaknesses:**

1. the singular stochastic sampling strategy (introducing a single stochastic sampling step) seems to be a special case of the MixGRPO in Li et al. 2025. They insert a window of stochastic sampling steps when collecting data. When the window size decreases to one, MixGRPO reduces to the singular stochastic sampling strategy.
2. it is not clear why the multi-granularity advantage integration is beneficial. Intuitively, the training and testing discretization steps should be consistent to ensure better performance.
3. The numbers reported in Table 1 only shows marginal improvement compared to baseline methods.

**Questions:**

Could the author make a detailed comparison to MixGRPO and illustrate why the multi-granularity advantage integration may help?

---

### Official Review · Reviewer_xvs4 · 2025-11-01

**Soundness:** 2
**Presentation:** 2
**Contribution:** 2
**Rating:** 2
**Confidence:** 4

**Summary:**

The paper looks at the problem of text to image generation and uses a GRPO variant to fine-tune a large model. The authors discuss a couple of major issues faced by prior works that use RL to fine-tune these models and propose solutions for the same. First is that methods employing SDE to generate the flow at every time step suffer from proper credit assignment. Cumulative errors along the trajectory can lead to large number of poor samples that can affect GRPO training and it is difficult to attribute the reward to the correct step in the generation process. The authors propose a mixed approach with SDE only at one step and ODE on the other steps to correctly evaluate the different generation directions. Secondly, to improve the granularity in denoised images, they propose using a multi-granular advantage integration i.e. generating for different granularities. They evaluate their method against a couple of baselines on a text-to-image dataset.

**Strengths:**

(1) The issue of proper credit assignment is well identified. Performing SDE at every timestep in the denoising process can lead to most images being poor with poor rewards.

(2) Moreover the assignment of the credit now goes to the correct step. Assigning the same reward to all steps made it unclear which step in the denoising led to the incorrect direction and which step led to the correct one.

**Weaknesses:**

(1) The discussion on granularity is unclear. Why is it needed? Are other issues being injected into the denoising process by jumping over time steps?

(2) Mix-GRPO is mentioned in the related work as a method that also mixes SDE-ODE. How is the proposed method different from Mix-GRPO?

(3) Experiments are limited. There is one dataset with one base model and two baselines. Is the base model finetuned over a dataset (through SFT) or is it directly the pretrained model?

(4) In the results, the improvement over the baselines seems to be small. If the baselines suffer from improper credit assignment, the results should indicate that. Are the improvements statistically independent?

(5) Flow-GRPO is not tested against.

**Questions:**

(1) What do you mean by in-domain and out-of-domain? These are different metrics and not the text distributions? Can a different metric really be called an “out-of-domain” metric?

(2) What do DanceGRPO and MixGRPO do and how is your method different from them?

(3) In related work, why is multi-turn RL mentioned? There is no discussion about multi-turn anywhere else in the paper.

(4) You mention that PPO is sensitive to reward model inaccuracies, since GRPO is also using the same reward, why isn't it sensitive to the same?

(5) Performing only ODE converges to the sample from the desired distribution. So does sampling using SDE. Does mixing the two still lead to a sample from the desired distribution or does the mixing break the theoretical guarantees?

(6) In the algorithm, what about $t<k$?

---

### Note · Authors · 2025-11-13

**Comment:**

After careful consideration, we have decided to revise the manuscript entirely. I sincerely thank the reviewers and the AC PC for their efforts.

**Withdrawal Confirmation:**

I have read and agree with the venue's withdrawal policy on behalf of myself and my co-authors.